# Time-Series PV Hosting Capacity Assessment with Storage Deployment

**Magdalena Bartecka** [1,*] **, Grazia Barchi** [2] **and Józef Paska** [1]

[1] Institute of Electrical Power Engineering, Warsaw University of Technology, ul. Koszykowa 75, 00-662 Warsaw, Poland; jozef.paska@ee.pw.edu.pl
[2] Institute for Renewable Energy, Eurac Research, viale Druso 1, 39100 Bolzano, Italy; grazia.barchi@eurac.edu
[*] Correspondence: magdalena.bledzinska@ee.pw.edu.pl

**Abstract:** Europe aims to diversify energy sources and reduce greenhouse gas emissions. On this field, large PV power growth is observed that may cause problems in existing networks. This paper examines the impact of distributed PV systems on voltage quality in a low voltage feeder in terms of the European standard EN 50160. As the standard defines allowable percentage of violation during one week period, time-series analyses are done to assess PV hosting capacity. The simulations are conducted with 10-minute step and comprise variable load profiles based on Gaussian Mixture Model and PV profiles based on a distribution with experimentally obtained parameters. In addition, the outcomes are compared with "snapshot" simulations. Next, it is examined how energy storage utilization affects the hosting capacity. Several deployments of energy storages are presented with different number and capacity. In particular, a greedy algorithm is proposed to determine the sub-optimal energy storage deployment based on the voltage deviation minimization. The simulations show that time-series analyses in comparison with snapshot analyses give completely different results and change the level of PV hosting capacity. Moreover, incorrect energy storage capacity selection and location may cause even deterioration of power quality in electrical systems with high RES penetration.

**Keywords:** hosting capacity; photovoltaics; energy storage system; power quality; power flow analysis; time series analysis

## 1. Introduction

In the last few years, the increasing interest to reduce the greenhouse gas (GHG) emissions from fossil fuel utilization is significantly pushing the increase of the renewable energy sources (RES). In Europe, the share of renewable energy in final energy use has doubled since 2005, reaching 17.6% in 2017 and 18% in 2018 according to the European Environmental Agency (EEA) [1]. Among different RES systems, the photovoltaic (PV) technology plays a central role due to its modularity (it is possible to install a small system on the roof or large field), its efficiency (last PV modules are able to reach 20–25% of power conversion) and cost effectiveness. In the last ten years in the European Union, the grid-connected PV system installations increased from 11.3 GW at the end of 2008 to over 117 GW at the end of 2018 [2]. This trend will continue in the next years according to the market development to 2030 in order to achieve the emission reduction. The constant increase of PV system integration may affect the proper electrical grid operation causing issues such as voltage deviations, thermal violations, harmonics distortions, which lead to a general poor power quality situations [3,4] and problems in protection systems. These problems are particularly stressed if we consider the typical variable nature of RES generation, which is strictly dependent on the weather conditions [5,6]. The power quality issues introduced by RES should be a problem to both sides, producer or producer and consumer

(prosumer) and Distributed System Operator (DSO). Indeed, for the electricity suppliers, poor power quality may result in higher economic costs such as refunds or fines. For final customers, poor power quality may affect the proper work and lifetime of several devices. For this reason, it is important to quantify the maximum power from PV that can be injected into the grid without affecting its reliability and efficiency. To this purpose, it is fundamental to assess the so-called hosting capacity (HC). There is not a unique method or solution for HC assessment because it really depends on different parameters such as the grid under test, the behavior of the installed RES, the performance index taken into account and the standard limits. However, in [7], authors indicate three main methods of HC assessment: deterministic, stochastic, and time series. Deterministic analyses use fixed size and location of power sources as well as fixed grid parameters. Examples of this approach are presented in [8–10]. Stochastic analyses include uncertainty of PV units, such as size, location, number of PV systems, etc. [11,12]. In these approaches, time of observation is shrank to few hours in a day or simply snapshot simulations are done as an example of the worst case, when PV power is maximal and the demand is minimal. The HC is assessed based on the probabilistic function. In opposite, a time-series approach includes the time-variability of PV generation and load over a specified period with an appropriate time step [13,14]. Time-series is very often the abbreviation of quasi static time-series (QSTS) analysis that refers to steady-state operating conditions in an electrical grid, where no transients are observed. This kind of approach based on the production-consumption variability over the time tends to be more realistic in the presence of renewable generations due to its different behavior over the year. However, a long observation interval increases significantly the computational time. Thus, many studies provide new techniques to increase the computation efficiency of QSTS analyses [15,16], the time of computing is still long, and the analyses are often shortened to one day with 1-h resolution. An example of hourly one day period analysis is presented in [17]. A stochastic PV deployment with a time series approach is combined; however, the simulation are conducted for light load (the worst case) and a PV profile is the same for every energy source distributed over an entire grid. In [18], authors proposed a similar probabilistic methodology of PV HC assessment with use of a Monte Carlo method. They improved rooftop PV models by adding variability of every single generator unit. In [14], four different hourly profiles of PV generation are applied and three PV installation scenarios are investigated to assess PV hosting capacity (PV HC). In these works, the hosting capacity is assessed based on the single violation appearance. Thus, one violation during performed time period is enough to claim maximal PV penetration. A longer period of time—one week simulation with variable load and generation profiles is presented in [19]; however, the authors focus more on grid modeling than PV hosting capacity assessment in essence. In [20], the concept of dynamic hosting capacity is presented that is variable and time dependent. For this HC assessment, variable PV profiles and load profiles were applied.

In many works, not only hosting capacity is assessed, but simultaneously some methods to enhance it are proposed [21,22]. Reactive power control and power factor correction are the effective way of balancing voltage deviations [23,24]. On-load tap changers optimal control also may be a solution for overvoltage and can increase HC as reported in [25,26]. Active power curtailment techniques can contribute to HC enhancement by minimizing overloading and thermal problems; however, the economic aspects should be taken into account [27]. HC based on harmonic distortion constraints can be enhanced by active power filters [28,29] or passive filters [30]. Moreover, battery energy storage system (BESS) utilization is widely discussed [31–33] as a method to increase HC by reducing voltage deviation and improving energy balance. In literature, it is commonly assumed that BESS in grids with high RES penetration are sited either as a central storage [34–36] or distributed units connected to energy sources [37,38]. In [39], these two arrangements were compared. The authors concluded that both solutions allowed the grid with high PV penetration to operate satisfactorily; however, the option with distributed BESS is more efficient economically. In [36], authors developed the BESS sizing methodology in order to improve HC of the middle voltage grid with the HV/MV transformer. The BESS capacity was calculated for 2-year time series data in order

to avoid overloading caused by additional wind generation. The HC limit was assessed on the basis of the first hour of violation appearance and further was calculated as the percentage of time. At the end, BESS capacity optimization was conducted taking into account the correlation between acceptable violation probability and the point of "diminishing returns". In [37], the authors define an objective function as the minimization of voltage deviations and power losses by optimal choice of energy storage capacity, site, and number. They proposed mixed genetic algorithms and constraint programming to solve the problem. In [38], the genetic algorithm is used to optimal allocate and size distributed ESS. The objective function is voltage rise minimization as well. Moreover, in [40], the review of energy storage system expansion planing methods is conducted. The most common objective function is minimizing the cost of investment or/and maintenance. The solutions take into account storage and grid constraints.

The presented literature overview shows that there is still a need to study hosting capacity assessment methods, which assure good accuracy and similarity to real grid operation with simplicity of their application. The assessment of HC is of crucial importance for most of the DSO, who are obliged to comply with national energy law and sales contracts. They should anticipate to what extent they can guarantee energy security and ensure power quality with the least amount of work and money. Moreover, it can be seen that there is no single, clear and complete solution for energy storage utilization to mitigate the impact of RES. Still, optimal capacity, location, and number are the variables that should be investigated in detail. Within this context, the present work aims to show a different approach to deploy battery energy storage in order to improve the hosting capacity. In particular, a placement technique based on greedy approach is proposed and compared with other simple techniques. The hosting capacity is assessed using the time-series PV and load profiles over a typical week where the violation limits are inspired by restrictions of the European standard EN 50160.

The paper is structured as follows: in Section 2, the background of hosting capacity term and power quality issues are discussed. In Section 3, the methodology of research is presented. In Section 4, the results of simulations are presented and analyzed while, in Section 5, the conclusions are made.

## 2. Theoretical Background

As mentioned in the Introduction, the power injected from a PV in the grid can cause power quality issues. In this section, it is briefly recalled how the voltage deviates from its nominal value in the presence of RES and the indices considered for the HC assessment are presented. Generally speaking, power quality problems may originate from nonlinear or asymmetrical loads, unbalance of generation and demand, transients, malfunctions of electric equipment, faults or high power devices' operation. Concerning renewable energy sources, the problems that appear are root mean square (RMS) voltage changes. The phenomena may be explained with the equivalent circuit, presented in Figure 1 and the Ohm's law.

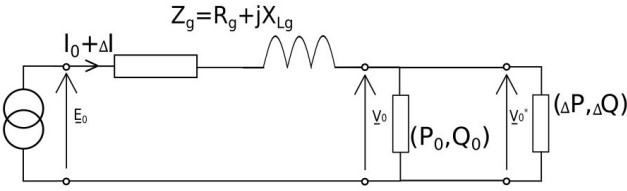

**Figure 1.** Grid equivalent circuit.

Voltage drop is calculated with use of Equation (1). Its value is dependent on the grid impedance, load type, and power. In the passive grids, the higher current flows through the line, and the higher voltage drop is observed. The same situation takes place when resistance or reactance of the grid rises. In a low voltage (LV) grid, long distance lines mean higher equivalent resistance and may lead to

constant undervoltage. For this reason, to assure a voltage level in required range all over the grid, the voltage in a substation is set on higher values than nominal [41].

However, in networks where consumers become producers (i.e., in presence of renewable energy generator), the distributed sources raise the voltage in the point of connection. Instead of voltage drop, the feeders are affected by overvoltage. Moreover, power from renewable energy sources is highly dependent on weather conditions, which may result with power and further voltage fluctuations. This change of demand may be expressed with Equation (2):

$$\frac{E_0 - V_0}{V_0} \approx \frac{\Delta V_0}{V_0} = \frac{R_g P_0 + X_g Q_0}{V_0^2} \tag{1}$$

$$\frac{\Delta V_0}{V_N} = \frac{V_0 - V_0^*}{V_N} = \frac{R_g \Delta P_0 + X_g \Delta Q_0}{V_N^2} \tag{2}$$

The hosting capacity due to renewable energy is defined as the maximal power produced by renewable energy sources, which will not affect both the supply reliability and electric power quality over required limits. This term is especially useful while installed green power constitutes a significant share of energy mix.

Hosting capacity is determined based on so-called grid performance indices. They concern many aspects, like work of protection equipment, voltage parameters, thermal insulation of cables, and transformers and others. The indices define the limits within which the safety grid operation is possible. In Europe, the boundary values for the medium and low voltage, as well as the duration of interruptions or the presence and amplitude of the harmonics components, are presented in the standard EN 50160 [42]. Three different types of voltage deviations are particularly considered:

- Voltage magnitude (LV, MV) violation of ±10% of the nominal value for 95% of the week, mean 10-min RMS values,
- Voltage fluctuations (LV)—5% normal, 10% infrequent,
- Imbalance (LV, MV)—up to 2% for 95% of week, mean 10 min, RMS values.

Although the standard comprises the supply voltage standard because of thermal issues, the focus is also on:

- Current (overcurrent which may lead to thermal issues),
- Maximal transformer rating.

Aforementioned indices are commonly the base to assess when maximum hosting capacity is reached. In our work, all of them are considered except from voltage fluctuations.

## 3. Methodology

### 3.1. Hosting Capacity Assessment

Usually, PV hosting capacity is assessed only for one type of simulation mode—snapshot or time-series. In this research, both of these approaches are compared. The flow chart to assess PV HC is illustrated in Figure 2 that has been inspired by [21]. The system is modeled, including the parameters of substations, lines, transformers, and coordinates. Then, receivers and PV systems are located in the same selected buses, and load power is determined. PV penetration is calculated with the use of Equation (3) [6], as it provides data about installed power that is easy to obtain for a DSO:

$$PV_\% = \frac{\sum_{n=1}^{N_{PV}} P_{PV_r}(n)}{\sum_{l=1}^{N_{load}} P_r(l)} * 100 \tag{3}$$

where: $n$ is the number of PV generators, $l$ is the number of buses with active power, $P_{PV_r}$ and $P_r$ refer to installed, rated power of PV, and load, respectively.

In [6], the research are conducted on real feeders. PV penetration range taken into account was 0–200%. This value is extended and PV penetration is changed in the range from 0 to 300% with a 10% step.

PV hosting capacity in static "snapshot" mode is calculated for fixed power values of PV systems and fixed power values of loads at a time point. The loadshapes are temporary disabled. In this analysis, constant single values are obtained. This approach is often applied in-line with "the worst case" scenarios, where PV power is peak and demand is minimal. To assess if the specified constraints are violated, the RMS voltage with allowable limits as well as power and current with maximal ratings are compared.

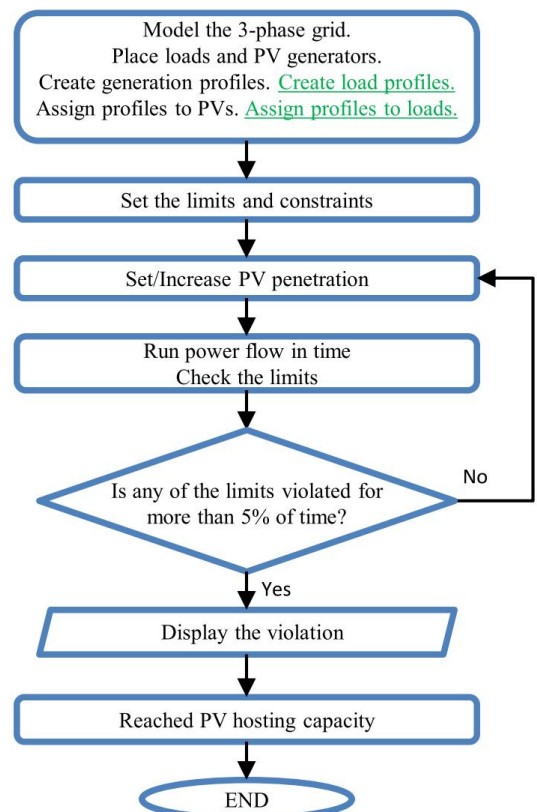

**Figure 2.** Flow chart of the algorithm to assess hosting capacity for system with PVs and loads (time-variable and constant).

To assess time-series PV HC, variations of PV generation and eventually load are included. Following the black font in the algorithm presented in Figure 2, HC is assessed for constant in time load and time-variable PV profiles. Added green font indicates the flow chart to assess HC for time-variable both PV profiles and load profiles. In both algorithms, system modeling is followed by creating typical time-variable profiles. For every PV system, an individual generation profile is created and eventually load profiles of active power and reactive power for every load individually are created. The time profile sets are normalized, so that the peak value during concerned period is equal to one, and other values are recalculated according to it. In brief words, for every PV penetration level, normalized load and PV profiles are not changed although their nominal power is modified. Profile normalization allows for easy comparison of result for increasing PV penetration levels and different simulation scenarios. In the next step, the constraints are defined. The power flow is run for one-week period and starts for 0% PV share. Hosting capacity is assessed based on the measurements gained during one week. The RMS values are measured every 10-min over this period. In every time

step, voltage, current in lines, and transformer power are compared with reference values of voltage threshold, line boundary current, and nominal apparent power of transformer, respectively. At every PV penetration level, the probabilities of violations are calculated. It has been mentioned before that, to acknowledge the violation, it needs to appear in more than 5% of 10-minute time segments for the set period (for one week, the maximal number of segments with violations is 51). Unless one of the constraint is violated for more than 5% of time, PV penetration is increased. However, if the results indicate that violations appear in more than 5% of samples, it is assessed as the PV hosting capacity is reached.

### 3.2. Energy Storage Placement

Once time-series hosting capacity is assessed, BESS have been added to the model, as the method to enhance hosting capacity. Among various scenarios of BESS location, a particular scenario based on a greedy suboptimal placement has been proposed. To assess PV HC for grid with storage, the flow chart presented in Figure 3 is followed. In all applied scenarios, both loads and PV units have previously created time-variable profiles implemented. For a particular PV penetration level, after calculating the probabilities of violations for grid with loads and PV systems only, the next step is BESS utilization. In every scenario, one storage at a time is added until their number equals the number of PV units. PV penetration level is increased as before in the range from 0 to 300% with 10% resolution. PV HC is assessed after the all cycle, based on the displayed violations.

The scenarios of storage deployments are as follows:

1. energy storages placed one by one from the closest to the furthest bus from the substation (ascending order) among buses with already installed PV units,
2. energy storages placed one by one from the furthest to the closest bus from the substation (descending order) among buses with already installed PV units,
3. energy storages placed randomly one by one among buses with already installed PV units, the values are averaged from 20 repetitions to obtain more credible results,
4. energy storages placed randomly one by one among all buses in the feeder, the values are averaged from 20 repetitions,
5. energy storages placed based on the greedy algorithm.

There are several motivations of such scenarios. The interconnection of PV and energy storages, used in the first three deployments, is considered the most common. Moreover, both the central and distributed BESS allocation is widely discussed in the literature as solutions to improve grid operation. In order to examine the impact of BESS utilization on the buses that are the most sensitive to voltage deviations, descending order scenario is applied. Finally, a greedy algorithm is implemented in order to find sub-optimal BESS distribution and compare the results with other solutions.

Greedy Algorithm for Storage Deployment

In the last part of this work, a greedy algorithm is used to find the sub-optimal BESS deployment. A similar approach was applied for measurement instrument placement in [43]. A greedy algorithm is a type of heuristic algorithm which finds a locally optimal solution at each stage with intention of finding global optimum. In every stage, it continues solving the problem based on the previously made decision. The algorithm not always yields to global optimum; however, it may approximate the solution within a reasonable amount of time compared to other optimization methods. In this work, the idea of greedy algorithm is followed for finding the best deployment of energy storages in the grid with high PV penetration. The objective function in every stage is the minimal voltage average root mean square error (Equations (4) and (5)). Based on the preliminary analysis of PV hosting capacity, it is observed that the most sensitive index is voltage level, thus it is applied. The objective function takes into account deviations from basic (primary) grid state without both PV units and storage:

$$f_{obj}(V) = ARMSE_V \longrightarrow min \tag{4}$$

$$ARMSE_V = \frac{\sum_{n=1}^{N_L} \sqrt{\frac{1}{K} \sum_{k=1}^{K} | V_{n,k} - \overline{V_{n,k}} |^2}}{N_L} \tag{5}$$

where $N_L$ indicates the number of lines, K number of time samples over one week period, while $V_{n,k}$ and $\overline{V_{n,k}}$ are the nominal (at primary state) and actual value of the interested quantity at line $n$, respectively. In simulations, it is assumed that the RMSE metrics will be applied to the absolute of complex bus voltage.

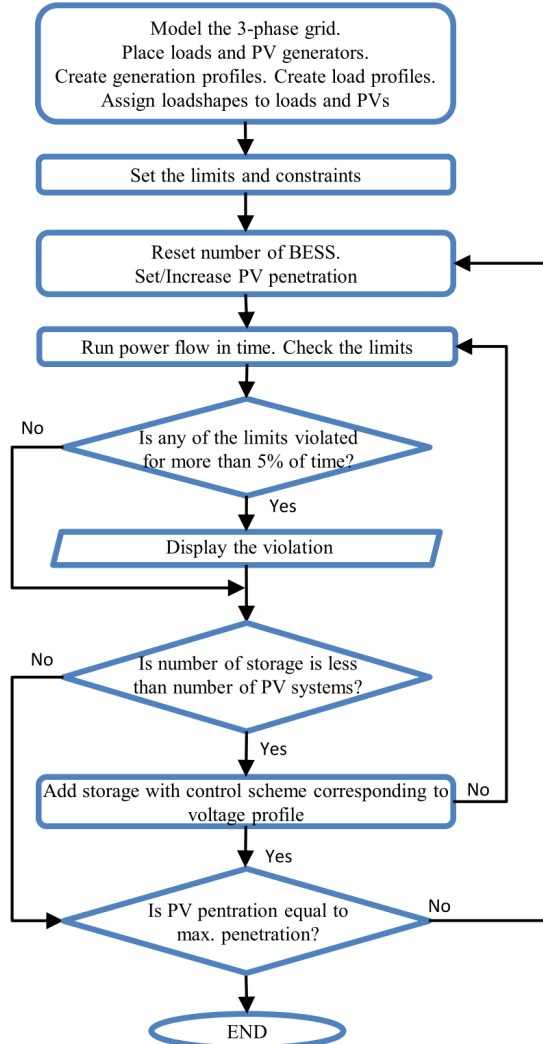

**Figure 3.** Flow chart of the algorithm to assess hosting capacity for the system with BESS.

## 4. Results and Discussion

For simulations, open source OpenDSS software is used, developed to carry out distribution grid analysis [44]. It enables computing in-time analysis for a three-phase network and assures high-precision results. The additional toolbox GridPV is used [45] via COM port, which for us is Matlab.

### 4.1. Grid Model

The feeder selected for the simulations is a real low voltage UK network, which has a typical radial structure [46]. Long distance lines lead to forks which then split into branches and supply load. With one voltage source and many receivers at the ends of the feeder, typically overcurrent is

observed at the main conducting line and undervoltage at the most remote nodes. The grid comprises of 131 buses. The feeder is modified by adding 30 loads randomly distributed, according to Figure 4. Load installed, rated power values (primary state) are randomly chosen within range 2–12 kW for a single load. The basic grid parameters are summarized in Table 1. More details may be found in [46].

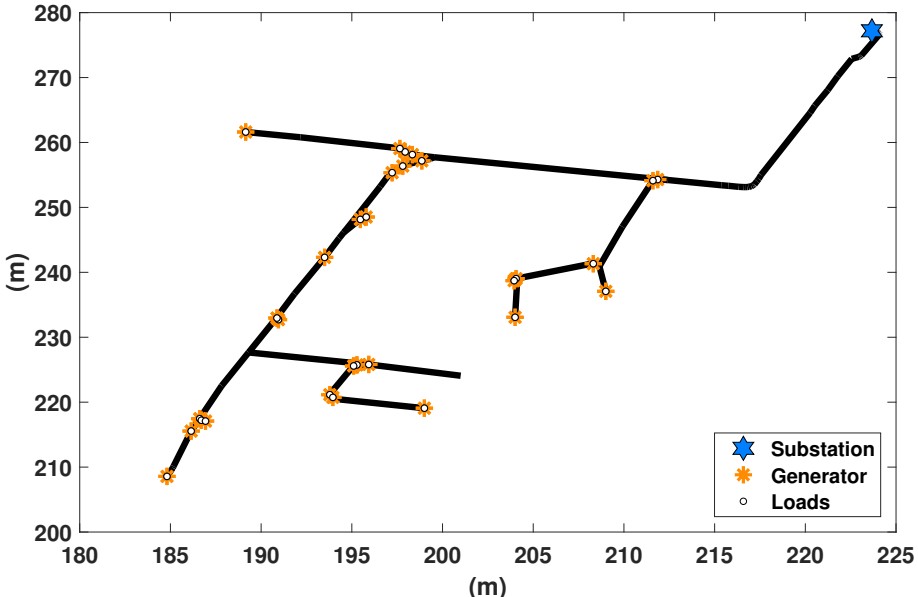

**Figure 4.** Graphical representation of the LV feeder under test with PV systems.

**Table 1.** Basic grid parameters.

| Element | Number | Characteristic |
|---|---|---|
| Transformer (Substation) | $N_{Tr} = 1$ | $S = 800$ kVA $n_V = 11/0.4$ kV |
| Buses | $N_b = 131$ | 3 terminals |
| Loads | $N_{loads} = 30$ | $P_r(l) \in \{2 - 12\}$ $PF = 0.98$ symmetrical, 3-phase |
| PV | $N_{PV} = 30$ | $P_{PV_r}(n)$ linked to $PV_\%$ symmetrical, 3-phase |

*4.2. Load and PV Profiles*

In the first part of the work, receivers have constant power in time. In the further steps, the fluctuations are imposed on them, resulting in the curve presented in Figure 5a). The green curve shows the normalized fluctuations of reactive power and blue of real power. Output instantaneous load power is a product of these curves multiplied by rated power and modified by power factor. The load profiles are created using the Gaussian Mixture Model distribution defining the number of peaks and the variances. A similar approach has been done by authors in [47]. Every load is symmetrical 3-phase and is characterized by an individual profile.

The PV normalized generation curves (called PV loadshape) are created on the basis of the stochastic distribution obtained by interpolating 10-minute data measurements collected in the experimental field of the airport of Bolzano (Italy) [47]. The output power is the product of one week profile and PV rated power. Figure 5b presents an example of the power curve of a three-phase PV generator. Every PV unit has an individual generation profile.

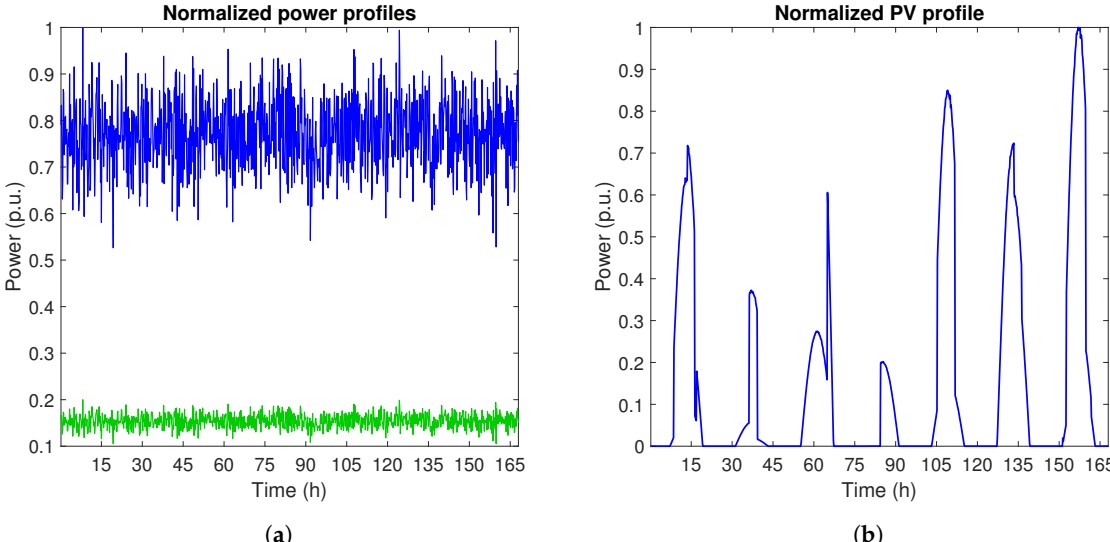

**Figure 5.** Example of active (blue) and reactive (green) load variations (**a**) and normalized PV production based on measured solar irradiance (**b**).

In present work, profiles have been created for one typical week, with no distinction between seasons. However, it could be seen from exemplary profiles that they include fluctuations that may reflect clouds, changes in irradiance, or load variability.

### 4.3. Energy Storage Model

Model of the storage in OpenDSS is a constant power injection with specified power factor. A storage may operate in three stages: idle, charging, and discharging. During charging, a storage power is negative. During discharging, the behavior is opposite and power is positive. In idle mode, a storage has some constant losses.

A simplified iterative algorithm was used to select the capacity and rated output power of BESS. Based on the preliminary results, RMS voltage is considered to be the most sensitive index. Thus, in order to minimize voltage fluctuations, the storage discharge/charge schedule was dependent on the actual voltage of the interconnected bus in the grid with specified PV share. Under/over a certain voltage level, the storage was triggered on/off in order to avoid voltage fluctuations over allowed limits. In the case that a storage state of charge was minimal or it was fully charged, then it went to idle mode. Following the concept of interconnection PV with energy storages, energy storages were placed in the same buses as PV units and loads. Once the control mode was given and energy storages were placed, time-series simulations with variable load and PV generation profile were conducted. For different PV penetration levels, storage power was changed iteratively in the range $P_{BESS} =< 0.1, 1.5 > \cdot P_{PV_r}$ [kW] and for every power value, storage capacity was changes in the range $C_{BESS} =< 1, 10 > \cdot P_{BESS}$ [kWh]. From all these combinations, the one with the lowest energy storage power and capacity for which the violation limits were not registered was selected. The results of optimal power and capacity differed for PV penetration levels. However, for higher PV penetration, the results were convergent and the sought storage power equaled 40% of a single PV unit power and capacity was 6 times higher. Since all the PV units were characterized by the same parameters, the expressions of total energy storage power and capacity are:

$$P_{BESS_{total}} = 0.4 * P_{PV_{total}} \qquad C_{BESS_{total}} = 6 * P_{BESS_{total}} \tag{6}$$

where $P_{PV_{total}} = \sum_{n=1}^{N_{PV}} P_{PV_r}(n)$.

Since the described method referred to the number of BESS equal to the number of PV generators ($N_{max}$), a single BESS power and capacity are obtained by dividing the values expressed by (6) by $N_{max}$.

In the further analyses, energy storage number differs in the range from 1 to $N_{max}$, thus a single energy storage power and capacity equal to:

$$P_{BESS_{single}} = \frac{P_{BESS_{total}}}{N_{BESS_{act}}} \qquad C_{BESS_{single}} = \frac{C_{BESS_{total}}}{N_{BESS_{act}}} \tag{7}$$

where $P_{BESS_{single}}$ indicates an individual BESS power and $C_{BESS_{single}}$ individual BESS capacity, $N_{BESS_{act}}$—the number of actually active storages ($< 1, N_{max} >$).

An example is one BESS ($N_{BESS_{act}} = 1$) of capacity equal to $C_{BESS_{total}}$ and power $P_{BESS_{total}}$ or 30 BESS ($N_{BESS_{act}} = 30$) of individual capacity equal to $C_{BESS_{total}}/30$ and power $P_{BESS_{total}}/30$.

### 4.4. PV Hosting Capacity Results

The first simulations include PV hosting capacity study in "snapshot mode"—constant, installed load rated power and constant, installed PV rated power. Rated loading of the grid is equal to 228 kW. In this case, neither maximal current nor apparent power constraints are violated that is shown in Table 2. Voltage is the most sensitive parameter; therefore, in further analyses, the focus is on it. Figure 6 illustrates voltage values across every line. It shows that PV hosting capacity is about 230%, while taking into account the limit of slow voltage changes (such as overvoltage) but two times lower—120%, while the limit of fast voltage changes would be considered. In the rest of the work, the voltage magnitude violations are considered to 5%. Due to the time-variations of loads and PV generation, it is assumed as a reasonable threshold to promptly detect voltage deviation.

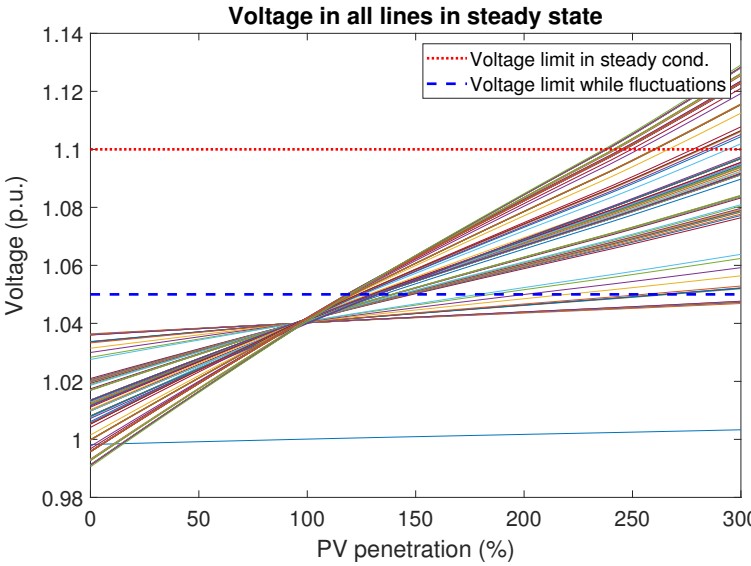

**Figure 6.** Voltage across all lines in the grid with rated loading in snapshot mode.

**Table 2.** System performance index violations in the examined PV penetration range.

|  | Voltage Magnitude | Voltage Imbalance | Current Rating | Transformer Rating |
|---|---|---|---|---|
| Snapshot | V | NV | NV | NV |
| Constant load | V | NV | NV | NV |
| Variable load | V | NV | NV | NV |

V—violated, NV—not violated.

Then, the simulations in the "snapshot mode" are done for another two different levels of loading, since it is expected that low loading of the grid causes more troubles in the field of voltage quality. PV penetration is calculated as previously, however, the actual power of every load is multiplied by 30% and 50% respectively in order to simulate "the worst cases". The results are presented in Table 3. Rated loading of the grid leads to HC at the level 120%. Load decrease by 50% causes HC drop to 70% while load decrease to 30% causes HC to drop to 50%.

Once the "snapshot" values are obtained, further simulations are conducted for one week period, including PV generation profiles and two load types—constant and variable. A constant load over time means that the load power is equal to the rated installed power during the entire simulation period. A variable load means that the rated installed power is only the base to be multiplied by the normalized load profiles. Thus, rated power may occur only in a few time steps during the simulation period. These simulations yield different outcomes of PV hosting capacity. For load set to 100% of rated, installed power, voltage imbalance, overcurrent, and power violation for any PV penetration level are not detected, which is summarized in Table 2. Voltage is the most sensitive index. A very slight difference is registered in phase shift compared to primary state. Figure 7 illustrates averaged voltage across all buses for 300% PV penetration during a week. It shows that some values of voltage in time series cross the upper threshold. In comparison, Figure 8 shows averaged voltage across all buses over a week the function of growing PV penetration. It rises linearly but does not cross the upper limit. As seen in Figure 7, diversified daily PV production causes voltage rise only during a few hours of a day, and these deviations do not significantly affect averaged values over one week.

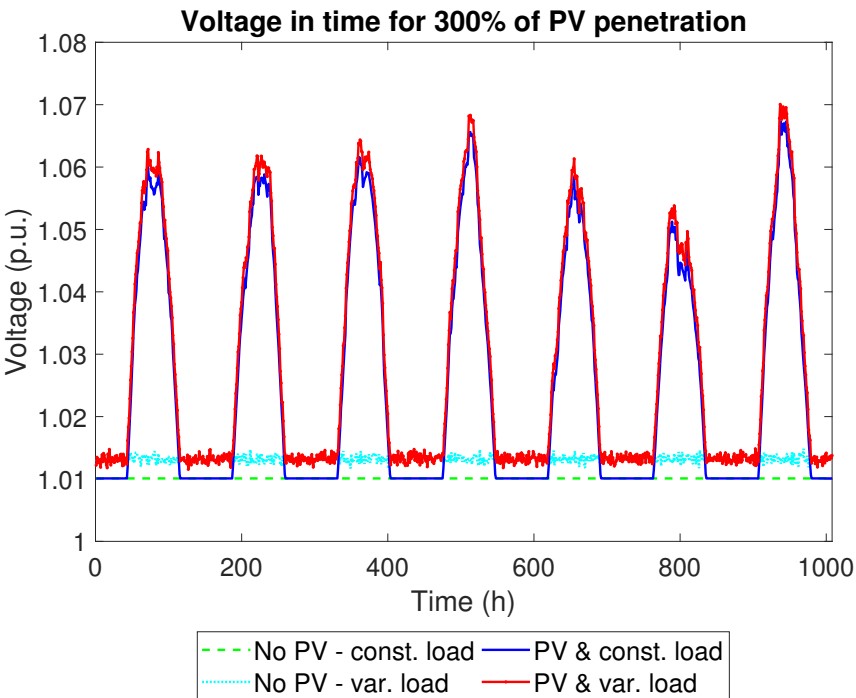

**Figure 7.** Averaged voltage across all buses for 300% PV penetration.

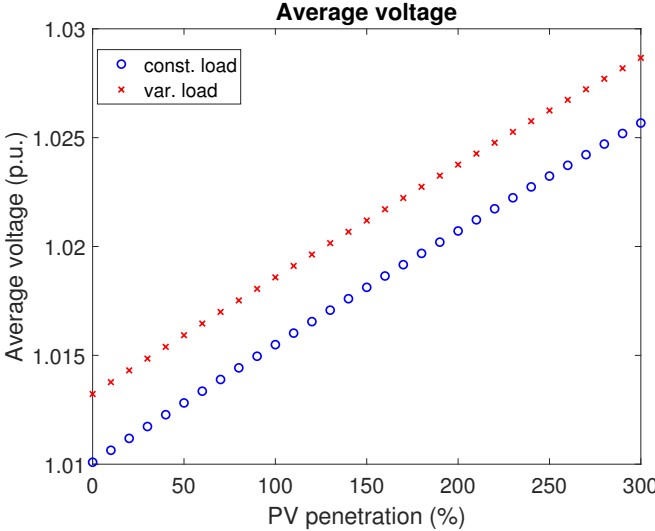

**Figure 8.** Averaged voltage across all buses and over a week.

For described cases, maximal PV penetration with respect to voltage limit 1.05 p.u. equals 200% for variable rated loading and 220% for constant rated loading. It can be explained in the following way. When the demand is constant over time, average loading is as rated during all measurement periods. Thereby, the grid is more loaded and more resistant to voltage rise. When the demand varies over time, rated power may occur randomly and rarely. Average total load power calculated for variable loading equals 93% of rated load. Thus, the worst case defined as maximal PV power and minimal demand is more probable to appear with variable loading resulting with PV hosting capacity decrease. Regarding deeper analysis of how the load decrease affects the hosting capacity, the simulations are done for another two different levels of actual total loading—30% and 50%. This is obtained by multiplying every single load rated power by the relevant factor. The simulations are conducted for both variable and constant load. PV penetration formula is not changed. Table 3 summarizes the results for PV hosting capacity for different simulation methods and different actual loading. Decrease by 50% rated loading leads to HC decline to 130% in a constant load case and to 120% in case of variable load. For 30% of rated loading, the HC is 80% and 70%, respectively.

**Table 3.** PV hosting capacity for different kinds of simulations and percentage of actual load.

| | **Actual Load** | **PV Hosting Capacity (%)** |
|---|---|---|
| Snapshot | $P_{act}(l) = 100\% \ P_r(l)$ | 120 |
| | $P_{act}(l) = 50\% \ P_r(l)$ | 70 |
| | $P_{act}(l) = 30\% \ P_r(l)$ | 50 |
| Constant load | $P_{act}(l) = 100\% \ P_r(l)$ | 220 |
| | $P_{act}(l) = 50\% \ P_r(l)$ | 130 |
| | $P_{act}(l) = 30\% \ P_r(l)$ | 80 |
| Variable load | $P_{act}(l) = 100\% \ P_r(l)$ | 200 |
| | $P_{act}(l) = 50\% \ P_r(l)$ | 120 |
| | $P_{act}(l) = 30\% \ P_r(l)$ | 70 |

The significant difference between time series HC and snapshot HC has two fundamental causes. First, for variable generation, every 100% PV penetration that corresponds to 100% of rated load power (228 kW) produces energy at the level of 29% of energy demand during the considered period. The average output PV power is ca. 62.85 kW, while the average load power is ca. 213.38 kW for variable load profiles. It yields to a higher imbalance in the relation load-generation. For snapshot mode, 100% PV penetration strictly corresponds to total rated power—the generation power is equal

to load power, and the energy balance converges to zero. The results of average output power of power sources and loads are summarized in Table 4. The second reason is that, for time-series mode, maximal PV power appears in a few time steps, and, if it is less frequent than 5% of a one-week period, the violation is not registered. These partly explain higher hosting capacity for time series simulations. Quite a bit difference is observed between snapshot and time-series analysis for all levels of load.

**Table 4.** Comparison of average output power of loads and PV units over a one-week period for three methods (100% PV penetration and nominal loading).

|  | Nominal Load | Avg. Load Power | Avg. PV Power | PV HC (%) |
|---|---|---|---|---|
| Snapshot | 100% $P_r(l)$ | 100% | 100% | 120 |
| Constant load | 100% $P_r(l)$ | 100% | 27.5% | 220 |
| Variable load | 100% $P_r(l)$ | 93% | 27.5% | 200 |

*4.5. Storage Implementation and Placement*

Based on the obtained results of time-series PV HC, a study of energy storage influence on the grid is carried out. Storage number varies from 1 to 30 for PV penetration in the range from 10% to 300% with 10% resolution. Several different deployments are applied and the effects are compared. As noted, the averaged voltage values over time are not sufficient in time series analysis. Therefore, further assessment is based on the event probability with respect to the norm EN 50160 [42] and the voltage deviation index from the primary state—ARMSE.

4.5.1. BESS Placed in Ascending Order

Storage units located in the buses with already installed PV units in ascending order from the closest to the furthest bus from the substation diminishes the probability of violations. First, violation is noticed at 280% PV penetration, and the event occurs for one high capacity BESS. When the number of BESS grows simultaneously with declining individual BESS capacity, the violations are not registered. At least three BESS are required to completely avoid violations in the examined range of PV penetration. Table 5 summarizes the results. Figure 9 presents voltage ARMSE metrics in the function of growing active storage number. It is observed that more storage units mitigate voltage deviations, although this trend is visible mostly for higher PV penetration. It is worth highlighting that a single storage of high capacity impacts the average voltage deviations more than many storages of lower capacity.

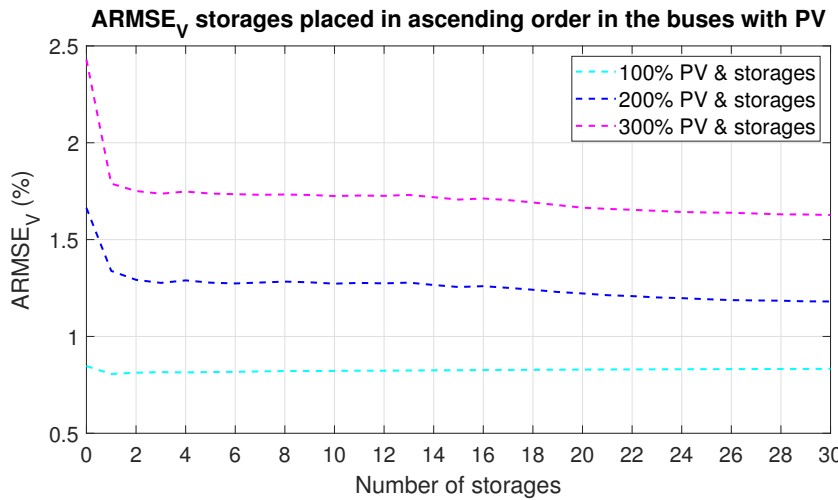

**Figure 9.** ARMSE of voltage across all buses and over a week for storages deployed in ascending order.

### 4.5.2. BESS Placed in Descending Order

The deployment of storages in descending order—from the furthest to the closest bus from the substation also decreases the power quality issues. Maximal PV penetration, for which the voltage violations occurred, is 250% for 1 high capacity BESS, but it concerned the lower voltage limit (0.95 p.u.). The event is caused by BESS control that was programmed to charge the BESS in the period of overproduction. The BESS charging schedule was implemented in the same way for all BESS deployment scenarios; however, in this case, it does not provide appropriate functionality. The upper voltage threshold is crossed for a higher PV share—280%. The grid operation is similar as in the case of BESS deployment in ascending order. The probability of crossing the voltage threshold dependent on the number of BESS is presented in Table 5. For a growing number of active BESS, PV hosting capacity increases and voltage ARMSE decreases, which is illustrated in Figure 10. There are slight differences between ARMSE metrics for ascending and descending BESS allocations.

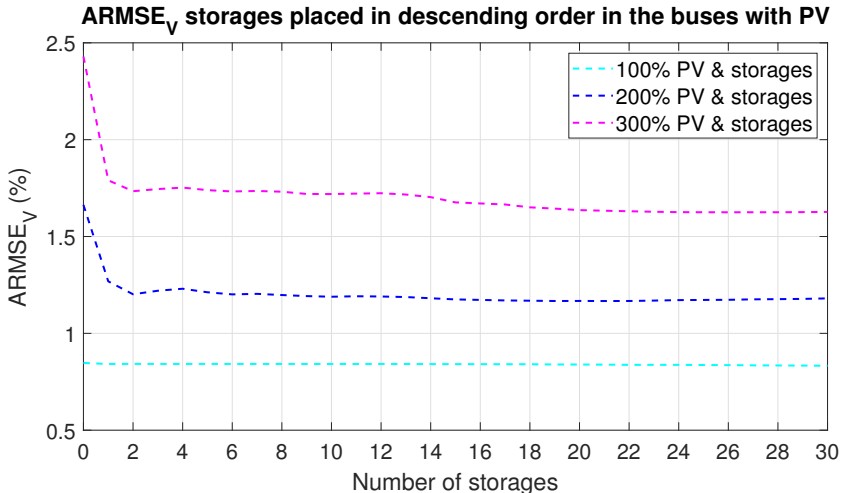

**Figure 10.** ARMSE of voltage across all buses and over a week for storages deployed in descending order.

### 4.5.3. BESS Placed in a Random Way among Buses with PV

Random distribution of BESS in the buses where PV units are installed proved that such allocation mitigates voltage quality issues. However, on the basis of averaged values from 20 repetitions, for 1 BESS of high capacity, maximal PV penetration is reached for 210% with probability of 5%. For further PV penetration growth, the probability rises. The increase of storage number alleviates voltage changes that are shown in Table 5. Figure 11 illustrates voltage ARMSE metrics in the function of a growing number of active BESS. It is worth highligting that this BESS placement gives one of the lowest HCs in comparison with different scenarios.

### 4.5.4. BESS Placed in Random Way among All Buses

Randomly allocated storages among all the buses in the grid (except the source bus) also increase PV hosting capacity. The worst effects again are observed for one storage of high capacity installed in the grid. From averaged results, for one active storage, maximal PV penetration is reached starting from 250% with 5% probability and it grows further. The growing number of BESS of lower capacity improves grid operation, which Table 5 shows. Voltage ARMSE metrics are presented in Figure 12.

Analyzing Figures 9–12 voltage deviation is the most visible for the highest PV penetration, while, for 100% PV penetration, the metrics remain almost unchanged for grid with and without BESS for all applied deployments. For such level of PV penetration, the produced and consumed energy is still balanced and does not affect grid operation significantly. In the study, BESS is programmed to react in the stages of excessive voltage rise/fall, which, in this case, does not occur very often. Thus, BESS

works for a long time in an idle state and its operation does not impact the grid (ARMSE metrics for 30 and 1 BESS hardly differ).

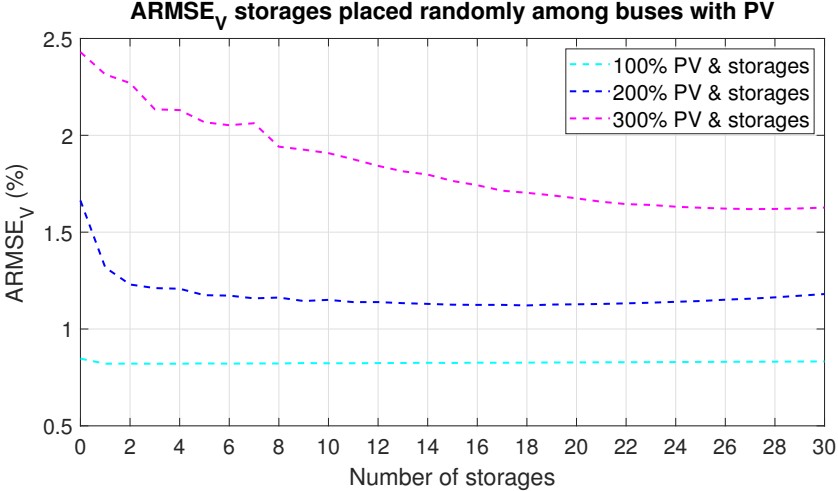

**Figure 11.** ARMSE of voltage over all buses and over a week for random distribution among nodes with PV units.

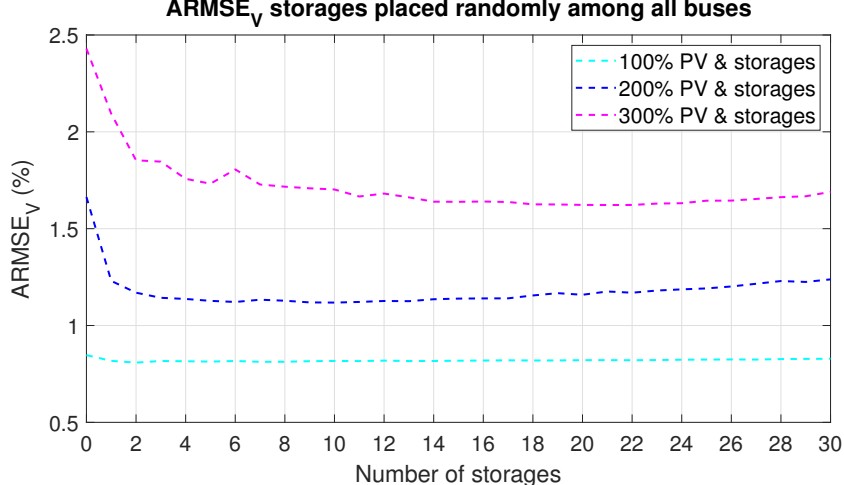

**Figure 12.** ARMSE of voltage across all buses and over a week for random distribution among all nodes.

### 4.5.5. BESS Placed on the Basis of Greedy Algorithm

The storage location based on greedy algorithm eliminates the probability of event appearance until 290% PV penetration. Figure 13 shows voltage ARMSE metrics. In comparison with previous results, a small decrease of the metric is observed. However, an interesting note is that the minimum is reached for one active BESS for 100% and 200% PV share and for two active BESS for 300% PV share. Because the algorithm finds a solution for every number of active storages anew, it is concluded that the relation between storage capacity and number is the best then. Another reason is the nature of the greedy algorithm that gives better solutions for a smaller number of optimization steps. It is worth mentioning that the buses with BESS, found as optimal solutions of greedy algorithms, do not converge with the buses where PV are installed. These observations confirm results obtained for BESS deployment among buses with already installed PV. Therefore, it should be noticed that this kind of storage allocation may not be the optimal solution for certain grids.

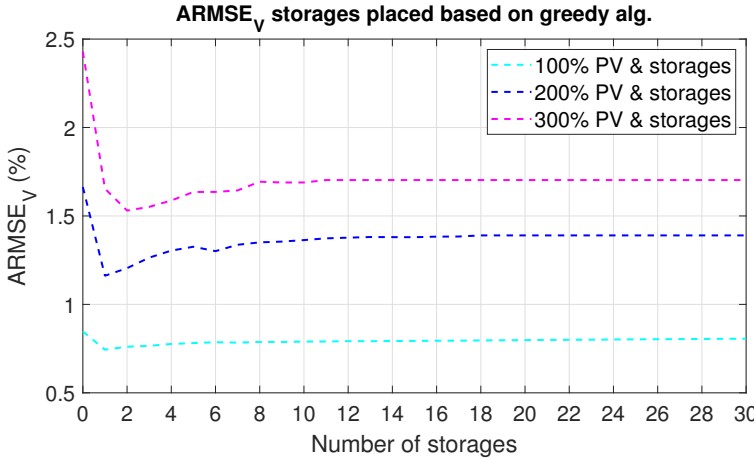

**Figure 13.** ARMSE of voltage across all buses and over a week for greedy-algorithm based storage deployment.

The comparison of voltage ARMSE metrics for all implemented storage deployments shows that the values for three allocations: ascending, descending, and random in buses with already installed PV units convergence to the same value for maximal number of storages (30). This proves correctness of obtained results. The smallest HC is noted for random BESS deployment among buses with already installed PV. In Table 6, the comparison of maximal PV penetration levels for investigated cases is summarized. Placing BESS in close distance to substations as well as based on greedy algorithms give the best results. Table 5 summarizes the voltage probability for every investigated BESS deployment for 300% PV penetration. In ascending, descending, and greedy BESS distributions, crossing voltage threshold enough times during one week qualifies the specific allocation to register the voltage probability or not, thus the values are 0 or 1. In the case of random deployment, the simulations were repeated 20 times to avoid misleading results. Hence, probability between 0 and 1 in the table refers to the number of the random distribution scenarios when the violations were registered (e.g., 0.05—in one from 20 random allocations, voltage violation was registered).

**Table 5.** Probability of voltage violation for 300% PV penetration and different BESS deployments.

| BESS Deployment | 1 BESS | 2 BESS | 3 BESS | 5 BESS | 10 BESS | 20 BESS |
|---|---|---|---|---|---|---|
| Ascending order | 1 | 1 | 1 | 1 | 1 | 0 |
| Descending order | 1 | 1 | 1 | 1 | 0 | 0 |
| Random among PV | 0.65 | 0.95 | 0.6 | 0.5 | 0.3 | 0 |
| Random among all | 0.7 | 0.3 | 0.3 | 0.05 | 0.05 | 0.05 |
| Greedy algorithm | 1 | 0 | 0 | 0 | 0 | 0 |

**Table 6.** Comparison of PV hosting capacity for different scenarios (with vs. without BESS).

| | | PV Penetration Limit (%) | |
|---|---|---|---|
| Snapshot | | 120 | |
| | Constant load | 220 | Only PV units |
| | Variable load | 200 | |
| Time analysis | Ascending order | ≥280 | |
| | Descending order | ≥260 (1 BESS) | |
| | Random among PV | ≥210 (1 BESS) | PV and BESS |
| | Random among all | ≥250 (1 BESS) | |
| | Greedy algorithm | ≥290 (1 BESS) | |

The proposed approach can be also extended to longer periods of time and can be differentiated according to the season.



This work shows the discrepancies between snapshot and time-variable based approaches to assess PV HC. Usage of stochastic methods, commonly investigated in the literature, based on the worst case scenarios may lead to an overestimation of the real HC. This approach is certainly safer, but it also assumes a 100% correlation between the two worst stages in the grid. In many works, this phenomenon is not even mentioned. In our study, differences between these two approaches are about 80–100%. Similar results were observed in [13]. In snapshot mode, HC reached because of overvoltage was 65%, while, for time-series simulations, none of the limits were crossed. In case of time-series simulations, the reactive power control and capacitor regulators mitigated the power quality problems.

One central BESS optimization to enhance HC by minimizing overloading have been previously reported in [36]. In [35], optimization of location and size of central BESS has been conducted in order to sustain voltage level in allowable limits. In comparison, simulations conducted by us show that one central BESS has worse performance than a few of smaller power and capacity when voltage deviation minimization is the objective function. The possible explanation is that distributed BESS can balance energy and voltage in distant nodes, improving in this way grid operation.

Genetic algorithm (GA) or hybrid GA for solving the problem of optimal BESS location and sizing are one of the most commonly developed [37,38]. Although they can assure good accuracy of the results, their implementation is quite complex. In comparison, an approach based on greedy algorithm, proposed in this paper, is a compromise between easy implementation and accuracy of the results. It provides HC enhancement by at least 10% in reference to other deployments for one central BESS unit. Increasing the number of BESS in this scenario eliminates violations in all examined PV ranges, while, for other BESS deployments, they still appear.

## 5. Conclusions

For selected grid model and configuration, the violations mainly affect voltage. A comparison of snapshot HC with time-series HC provided in this paper shows that time-variable profiles have great influence on the HC evaluation. With respect to the standard EN 50160, even very severe violations are allowable if 10-minute RMS values are within the accepted range in 95% of time during one week. Therefore, the variability of load and renewable generation unavoidable in real networks cannot be neglected while forecasting the grid operation with high renewable energy source shares. This could significantly increase the HC.

For the examined range of PV penetration and specified installation of PV generators, utilization of BESS improves grid operation. Thus, BESS may be the solution for growing RES shares. The research clearly indicates that fewer BESS of higher capacity is worse than few of lower capacity from the power quality side. It is especially visible in event probability evaluation.

Random BESS deployment among buses with already installed PV gives interesting results. HC has increased by 10% in comparison with HC obtained for a grid without BESS. This deployment turned out to have the poorest performance. It is a little surprising considering the fundamental BESS application of RES balancing, with which they are very often directly linked. Therefore, when BESS are applied to balance voltage deviations in a larger area of a grid, the proximity of RES should not be the priority while selecting storage locations.

The greedy algorithm for storage deployment assures slight improvement of ARMSE metrics in comparison with other scenarios, along with PV HC improvement. It also seems to be more reliable for a lower number of BESS. Moreover, it has to be admitted that using Matlab combined with OpenDSS for these simulations is very time-consuming and some bottle-necks may be observed with computing efficiency. The metrics used in an objective function (votlage ARMSE) of a greedy algorithm is probably not the most suitable, as it provides cumulative value and does not report about individual voltage violation.

Based on the research, it is concluded that storage charging/discharging scheme as well as capacity and rated power are of great importance while facing grid balancing. Incorrect selection of these parameters may lead even to deterioration of power quality instead of improving.

The proposed approach for PV hosting capacity assessment and BESS influence is practical and simple enough to be implemented in the process of grid planning. It is universal and may be applied for different grid topologies and resource data. Especially nowadays, when many meteorological databases are available to the public and smart metering structures collect data about loading, long-time simulations may reflect in a good way the real grid behavior.

**Author Contributions:** Conceptualization, M.B. and G.B.; methodology, M.B. and G.B.; software, M.B. and G.B.; validation, M.B., G.B., and J.P.; formal analysis, M.B., G.B., and J.P.; investigation, M.B. and G.B.; resources, M.B. and G.B. and J.P.; data curation, G.B. and M.B.; writing—original draft preparation, M.B.; writing—review and editing, G.B. and J.P.; visualization, M.B. and G.B.; supervision, G.B. and J.P. All authors have read and agreed to the published version of the manuscript.

**Funding:** This research received no external funding

**Conflicts of Interest:** The authors declare no conflict of interest.

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
