# Peer review of "Time-Series PV Hosting Capacity Assessment with Storage Deployment"

_energies, doi:10.3390/en13102524_

Round 1

Reviewer 1 Report

Manuscript aim is interesting for the development of RES , basically wind and photovoltaic. Nevertheless, some considerations of the application of EN 50160 to PV plants with BESS should be more detailed.

It would be also important to clarify the selected week profile.For instance, the days for one week in summer frequently are clear, but in spring often the weather could be very variable (sunny / partial cloudy /cloudy). For this reason, results could be different. In the same way, load profile could be highly variable. According to EN 50160, how many one-weeks periods should be analysed to establish normal operating conditions?.

Please, justify why results shown in eqs. 4 are the same for differents load profiles and PV power profiles during the period analysed (one week). Are the same for another periods?.

In figure 4, the normalized PV profile is for 1000 h?. How and why has been normalized based on measured solar irradiance for a week?.

Some nomenclature should be defined (include DSO in line 28, used in line 75). Low voltaje (LV) in line 121,… In line 58 grid intead of gird.

Reviewer 2 Report

In order to solve problem analysed in this paper it is desireble to mention active power filters and power factor correctors. Performance of these devices improve power quality in electrical netwoks.

Analysis of electric network with  4 element network  is important for basic understanding, however modern electrical networks involve lot of electronics, eg: https://www.hindawi.com/journals/ijp/2011/643912/

or similar:

https://ieeexplore.ieee.org/document/7754921

https://link.springer.com/chapter/10.1007/978-3-319-29357-8_18

https://ieeexplore.ieee.org/document/964762

One paper writen by KM Smedley et all investigates position of the same active power filters in order to obtain highest stability of electrical network https://academictree.org/etree/publications.php?pid=456383

This issue at least should be mentioned.

My sugestion is to add few references more.

Reviewer 3 Report

Dear authors, I have read with interest your work entitled "Time-series PV hosting capacity assessment with storage deployment", and while recognizing the validity of the work done, I do not believe that it can be published in its current form, and I recommend that it be restructured.

One important shortcoming is that the contribution to the state of the art of the work presented is not clearly outlined. As it is presented, the work appears to be an application of methodologies and tools already available to a hypothetical scenario. I believe that it can be structured in a clearer way by exposing the contribution to the state of the art and by clearly explaining the methodologies of investigation, also in order to evaluate the completeness of the research carried out, its accuracy, and the validity of the conclusions drawn.

The exposition of the text should be reviewed because in several points, and in particular in the first part of the introduction, the style is very fragmented, difficult and not pleasant to read.

The methodology is not clearly exposed, the lack of a clear statement about the research methodology applied does not allow to follow, in an easy way, the different simulations made, which follow a logical thread but for which it is difficult to assess completeness and correctness. The algorithms used are not described exhaustively, and the necessary references to the calculation tools used are totally missing.

The results are presented in a very detailed but discursive form (use tables plots and diagrams when possible), the overall view of the results obtained is not clear and the discussion lacks an accurate comparison with the state of the art that would allow a better evaluation of the results.

For these reasons, I suggest resubmitting the work after a major revision.

Round 2

Reviewer 1 Report

The authors have made all the corrections requested and the paper has improved in quality so my suggestion is to Accept the paper for publication in Energies Journal.

Author Response

Thank you for the positvie review.

Reviewer 3 Report

Dear authors, I think you have done a very good revision work. The text is much clearer to read, making it easier to understand the results obtained. All objections to the first draft have been very well resolved. I am in favour of publication.

I would like to mention two minor notes:

line 237: please insert a citation (or the urls in a footnote) for opendss and gridpv

In figure 5 please specify the colors used for active and reactive power in the caption.

Author Response

Thank you for the positive feedback. Two additional citations have been added  (line 236) and the caption in Figure 5 has been completed.